

# Transferability of ML-based Global Calibration Models for NO$_2$ and NO Low-Cost Sensors

Ayah Abu-Hani[1], Jia Chen[1], Vigneshkumar Balamurugan[1], Adrian Wenzel[1], and Alessandro Bigi[2]

[1]Environmental Sensing and Modeling, Technical University of Munich (TUM), Munich, Germany
[2]Department of Engineering "Enzo Ferrari", University of Modena and Reggio Emilia, Modena, Italy

**Correspondence:** Jia Chen (jia.chen@tum.de)

**Abstract.** It is essential to accurately assess and verify the effects of air pollution on human health and the environment in order to develop effective mitigation strategies. More accurate analysis of air pollution can be achieved by utilizing a higher-density sensor network. In recent studies, the implementation of low-cost sensors has demonstrated their capability to quantify air pollution at a high spatial resolution, alleviating the problem of coarse spatial measurements associated with conventional

monitoring stations. However, the reliability of such sensors is in question due to concerns about the quality and accuracy of their data. In response to these concerns, active research efforts have focused on leveraging machine learning (ML) techniques in the calibration process of low-cost sensors. These efforts demonstrate promising results for automatic calibration, which would significantly reduce the efforts and costs of traditional calibration methods and boost the low-cost sensors' performance.

    As a contribution to this promising research field, this study aims to investigate the calibration transferability between

identical low-cost sensor units (SUs) for NO$_2$ and NO using ML-based global models. Global models would further reduce calibration efforts and costs by eliminating the need for individual calibrations, especially when utilizing networks of tens or hundreds of low-cost sensors. This study employed a dataset acquired from four SUs that were located across three distinct locations within Switzerland. We also propose utilizing O$_3$ measurements obtained from available nearby reference stations to address the cross-sensitivity effect. This strategy aims to enhance model accuracy as most electrochemical NO$_2$ and NO sensors

are extremely cross-sensitive to O$_3$. The results of this study show excellent calibration transferability between SUs located at the same site (Case A), with the average model performance being of $R^2 = 0.90 \pm 0.05$ and RMSE = $3.4 \pm 0.9$ ppb for NO$_2$, and $R^2 = 0.97 \pm 0.02$ and RMSE = $3.1 \pm 0.8$ ppb for NO. There is also relatively good transferability between SUs deployed at different sites (Case B), with the average performance for NO$_2$ being $R^2 = 0.65 \pm 0.08$ and RMSE = $5.5 \pm 0.4$ ppb, and $R^2 = 0.82 \pm 0.05$ and RMSE = $5.8 \pm 0.8$ ppb for NO. Interestingly, the results illustrate a substantial improvement in the calibration

models when integrating O$_3$ measurements, which is more pronounced when SUs are situated in regions characterized by elevated O$_3$ concentrations. Although the findings of this study are based on a specific type of sensor and sensor model, the methodology is flexible and can be applied to other low-cost sensors with different target pollutants and sensing technologies. Furthermore, this study highlights the significance of leveraging publicly available data sources to promote the reliability of low-cost air quality sensors.



## 1 Introduction

Interest in air quality (AQ) has increased significantly over the past decades as a result of the severe impact of air pollution on the environment and public health (WHO, 2004). Major air pollutants such as carbon monoxide (CO), nitric oxide and nitrogen dioxide (NO + $NO_2$ = $NO_x$), particulate matter (PM), and anthropogenic Volatile Organic Compounds (VOCs) originate mainly by anthropogenic activities that directly and indirectly affect the AQ and public health (Kelly and Fussell, 2015). Consequently, monitoring and mitigating air pollution is of utmost importance in support of sustainable development. To date, the official regulatory monitoring stations use high-precision instruments based on optical measurement principles (e.g. in the chemiluminescence method in case of $NO_2$) that are highly cost intensive. The unit price for a fully equipped regulatory monitoring station varies from €50,000 to €100,000, in addition to maintenance and operating costs (Mead et al., 2013; Ionascu et al., 2021). According to the current European Ambient Air Quality Directive (2008/50/EC), implemented by EU member states, the microscale siting of a monitoring station for atmospheric pollutants subject to regulatory limits has several requirements. Among these, the site should be within 10 m from the road edge, and at least 25 m from high-traffic intersections. These high costs and space requirements constrain their spatial distribution to few areas. Moreover, as shown by several studies (e.g. Zhu et al. (2020); Beckwith et al. (2019); Baruah et al. (2023)), $NO_2$ hot spots at urban sites are not fully represented by their corresponding monitoring station.In order to bridge the gap, it's crucial to increase the spatial coverage of air quality monitoring. A possible way to do this is by using networks of low cost sensors along with modeling.

The drive to promote spatial coverage of air quality monitoring, combined with advancements in sensor technology, has paved the way for the utilization of low-cost sensors in air quality monitoring (Ionascu et al., 2021). Due to their affordability, portability and simple deployment, utilization of low-cost sensors have been widely acknowledged (Karagulian et al., 2019; Suriano and Penza, 2022; Snyder et al., 2013; Bigi et al., 2018). However, concerns about the stability of their performance and the quality of the data have significantly reduced their implementation on a large scale. Low-cost sensors for gas detection are mostly metal oxide and electrochemical sensors (Spinelle et al., 2015; Borrego et al., 2016; Mijling et al., 2018) and when deployed in environmental conditions, they suffer from drift, cross-sensitivity, and induced bias dependent on relative humidity or temperature (Masson et al., 2015; Mueller et al., 2017; Maag et al., 2018; Tagle et al., 2020; Papaconstantinou et al., 2023). This type of sensors are generally subject to two main sources of error: internal errors arising from the sensor's working principle and external errors resulting from environmental factors. Internal errors include variable detection limits, drift, and non-linear response. Identical sensors can introduce bias even when deployed at the same site, mainly due to manufacturing tolerances. External errors are mainly attributed to environmental factors such as temperature and relative humidity, as well as cross-sensitivity to interference gases (Ionascu et al., 2021; Giordano et al., 2021). In response to certain temperatures or relative humidity levels or changes in their values, low-cost sensors can exhibit significant biases. For example, both Masson et al. (2015) and Tagle et al. (2020) reported such high biases for $NO_2$ electrochemical cells during periods of high relative humidity (above 75 %). Widely used $NO_2$ electrochemical cells have been shown to have significant cross-sensitivity to $O_3$ (Miech et al., 2021; Spinelle et al., 2017; Alphasense Ltd, 2022). As a solution, an $O_3$ scrubber was added (Hossain et al., 2016; Alphasense Ltd, 2022) and it was shown that the filter material was successful at removing $O_3$ without affecting the signal due





to the target $NO_2$. Notwithstanding this scrubber, $NO_2$ cells still show some $O_3$ intereference. For example, according to Miech

et al. (2021), Alphasense $NO_2$-B43F exhibits 6.6 % cross-sensitivity to $O_3$, which also increases with time, as indicated by Spinelle et al. (2017). As a result, this interference induces a bias in the response.

Low-cost sensors biases can be partially mitigated through calibration, usually performed either under laboratory controlled conditions or by field co-location next to a reference monitoring station (Miech et al., 2021). Studies show that the latter approach is more satisfying and commonly used as it maximizes the performance of such sensors in real-world applications

(Spinelle et al., 2017; Suriano and Penza, 2022; Kureshi et al., 2022). Successful calibration has the potential to significantly enhance the AQ measurement process and reduce overall costs (Zimmerman et al., 2018; Munir et al., 2019; Van Zoest et al., 2019). However, the type and amount of processing applied to the air quality sensor data can lead to confusion about whether the processed data remains a true sensor measurement or a blend of secondary data and predictions. To address this issue, Schneider et al. (2019) proposed a standardized terminology for processing levels of air quality low-cost sensor systems. A 4-

level sequence ranges from Level-0 (raw sensor output) to Level-4 (processed data with spatial interpolation or assimilation into models). Each level serves different purposes, and data usability varies depending on the application. The proposed terminology aims to enhance the use and understanding of this technology and to ensure that the methods applied are well-documented and fit for their intended purpose.

Several calibration techniques have been reported in the literature, spanning from environmental factor correction (Miech

et al., 2021; Van Zoest et al., 2019; Kim et al., 2018), simple linear regression models (Okorn and Hannigan, 2021) to machine learning (ML) techniques (Nowack et al., 2021; Bigi et al., 2018; Zimmerman et al., 2018; Spinelle et al., 2015; Ionascu et al., 2021). Although some low-cost sensors outputs show an approximately linear relationship with the target pollutant, this linearity varies with time due to sensor aging (Li et al., 2021). ML algorithms have shown superior ability to intrepret such complexity of low-cost sensors, especially when including covariates that account for meteorological and environmental

variability. One of the most popular ML algorithms is Random Forest, which is an ensemble algorithm based on decision trees (Breiman, 2001). Random Forest, in addition to other commonly used methods such as Multiple Linear Regression, Support Vector Regression and Artificial Neural Networks, have been widely employed in air quality low-cost sensors calibration, and in some aspects of atmospheric chemistry, as they tend to outperform linear regression models (Nowack et al., 2021; Bigi et al., 2018; Zimmerman et al., 2018; Spinelle et al., 2015; Ionascu et al., 2021). Most studies available in the literature investigate the

individual localized calibration approach, in which, a single calibration model is created for each sensor unit (SU) after being co-located with a reference instrument (Zimmerman et al., 2018; Spinelle et al., 2015). Recent works, such as Bigi et al. (2018); Sahu et al. (2021); Van Zoest et al. (2019) and Nowack et al. (2021), studied individual calibration models considering site transferability, where they investigated whether a co-location-based calibration at one location produces reliable measurements at a different location. Bigi et al. (2018) found a performance range of about 6.5 ppb root mean square error (RMSE) for $NO_2$

and NO.

Only a few studies consider the calibration transferability (global calibration) among different sensors of the same make, including site transferability. A study conducted by Malings et al. (2019) evaluated the performance of individualized calibration models versus generalized calibration models. Individualized models are built based on data from a single sensor, while gener-





alized models combine data from all sensors of the same type. The researchers found that the most effective calibration model

type varied by sensor technology; for example, simpler regression models produced the best results for electrochemical CO sensors, while more complex models, such as Artificial Neural Networks and Random Forest models, provided the best results for $NO_2$ sensors. Although the outcomes varied, it was found that generalized models performed better at new locations compared with individualized models, despite slightly lower performance during initial calibration. Vikram et al. (2019) proposed a method for improving calibration transfer of $NO_2$ and $O_3$ by training calibration models on multiple sites. They rotated nine

SUs among three sites with reference monitors and introduced a novel split-NN approach which incorporates two sets of models: a global calibration model that combines data from a set of similar sensors spread across different training environments and sensor-specific calibration models that correct the sensor-to-sensor variations. The approach demonstrates versatility, accommodating linear regressors (LR) or NN for sensor-specific models and utilizing a two-layer NN for global calibration. The researchers found that the split-NN method performed better than Random Forest, reducing errors by 0%-11% for $NO_2$ and

6% -13% for $O_3$. In case of training their models on two sites and testing it on a third site with no overlap between the training and test data distributions ("Level2" benchmark as classified by Schneider et al. (2019)) resulted in a RMSE between 6 and 8 ppb for $NO_2$. Another study by Okorn and Hannigan (2021) examined the transferability of simple LR calibration models between several metal-oxide sensor systems (pods), focusing on ozone and methane. In their study, calibration transferability was performed among pods within the same location (i.e., sensors here share the same environmental variability). They sug-

gested using a standardization approach to normalize sensor signals for enhanced calibration transferability among units. A recent study by Wang et al. (2023) examined the calibration transfer performance of five low-cost SUs for PM and $NO_2$. The five SUs were collocated with a reference-grade monitor at one site for four weeks, and then two units were transferred to another site for a 16-day mobile campaign six months after the first deployment. The results show transferability between SUs located at the same site (same stationary settings), with the coefficient of determination ($R^2$) of best performing calibration

models for PM exceeding 0.80, and with $R^2$ for $NO_2$ units ranging around 0.70. However, models trained in stationary settings are difficult to transfer to mobile settings with different environmental characteristics.

    In our study, we developed global ML-based calibration models for electrochemical cells targeting $NO_2$ and NO, using data of low-cost SUs that were utilized in a previous study by Bigi et al. (2018). We focus on calibration transferability among SUs when deployed at the same location (i.e., same environmental characteristics) and different locations (i.e., different

environmental characteristics), given that no explicit overlap exists between the training and testing data distributions. This approach uses simple standardization to account for sensor-to-sensor variations, unlike the approach proposed by Vikram et al. (2019), which utilizes a ML-based method. In addition, this study presents potential improvements to model transferability by using additional information ($O_3$) from nearby regulatory air quality monitoring stations. This approach assists in untangling the interference of $O_3$ that persists in the $NO_2$ cells despite the presence of an an $O_3$ scrubber. (Spinelle et al., 2017; Miech

et al., 2021; Li et al., 2021). The incorporation of information from nearby regulatory monitoring stations is referred to as Level-3 in the classification by Schneider et al. (2019). Finally this study provides an opportunity to study the influence of geographical and seasonal variations on calibration transferability.



In Sect. 2, the sensor units, deployment sites and calibration methods are described. Results and discussion are found in Sect. 3. Finally, the main conclusions are drawn in Sect. 4. All data processing was performed with MATLAB (MathWorks, Natick, MA, USA) version R2021b.

## 2 Materials and Methods

### 2.1 Sensor units

This study utilized data collected from four SUs developed jointly by Empa, the Swiss Federal Laboratories for Materials Science and Technology, and Decentlab GmbH. These SUs were described and employed in previous studies (Bigi et al., 2018; Kim et al., 2018). Each SU consists of four electrochemical sensors: two $NO_2$ sensors (Alphasense NO2-B43F) and two NO sensors (Alphasense NO-B4), along with temperature (T) and relative humidity (RH) sensors (Sensirion STH21). All signals were sampled every 20 s, aggregated to a 1 min mean value, and transmitted to a central database every 180 min. The four SUs are denoted as AC009, AC010, AC011, and AC012, and the electrochemical sensors are denoted as NO_A, NO_B, NO2_A, and NO2_B, provided in millivolt. Throughout this study, signals of each electrochemical sensor represent the voltage difference between the working (we) and auxiliary (aux) electrodes. Data collected from the SUs and their corresponding reference instruments were preprocessed for outlier removal, smoothing, and averaging over 10 min, following the same procedure explained in Bigi et al. (2018).

### 2.2 Deployment sites and co-location

Over a two-phase campaign, SUs were deployed at three locations representing different emission and meteorological conditions in continuous co-location at quality regulatory stations within the National Air Pollution Monitoring Network (NABEL) (Bigi et al., 2018). A detailed description of the two-phase campaign can be found in Table 1. The first phase began in April 2017 and lasted for approximately three months, during which the four SUs were installed in the rural site of Härkingen (HAE), facing a major highway. This peculiar location allowed sensors to be exposed to both traffic-related pollutants, as the southern wind carries polluted air from the highway, and cleaner air masses, as the northern wind flows over the rural area. After the first phase of the campaign was accomplished, the SUs were transferred to two different locations: AC009 and AC010 were installed in Zurich-Kaserne (ZUE), while AC011 and AC012 were installed in Lausanne (LAU). The second phase lasted for around four months (from 28th July – 5th December 2017). All reference instruments provide measurements for NO, $NO_2$, $O_3$, temperature, and relative humidity. Fig. 1 summarizes the meteorological variables and pollutant concentrations at the different deployment sites, as measured by the reference instruments. In the vicinity of co-location site ZUE, there are four other nearby regulatory air quality monitoring stations located within an approximately 2.7 km radius. In Lausanne, there are two nearby stations situated within a radius of about 10.7 km from the co-location site (LAU), while none is available in Härkingen, see Fig. 2. These nearby monitoring stations provide $O_3$ measurements that are used to assess the potential enhancement of





**Table 1.** Details of the two-phase campaign of SUs deployments.

| Deployment Site | | Site Characteristics | Sensor Unit | Sample Size | Deployment / Co-location Period | Site Coordinates |
|---|---|---|---|---|---|---|
| First Deployment | Härkingen (HAE) | Rural & highway air masses (wide range of pollutants concentrations) | AC009 | 13478 | 13 Apr 2017 - 20 Jul 2017 | 47.311°N 7.820°E |
| | | | AC010 | 10202 | 5 May 2017 - 20 Jul 2017 | |
| | | | AC011 | 13478 | 13 Apr 2017 - 20 Jul 2017 | |
| | | | AC012 | 13478 | 13 Apr 2017 - 20 Jul 2017 | |
| Second Deployment | Zurich (ZUE) | Urban - background | AC009 | 18200 | 28 Jul 2017 - 5 Dec 2017 | 47.378°N 8.530°E |
| | | | AC010 | 18200 | | |
| | Lausanne (LAU) | Urban - traffic | AC011 | 18854 | | 46.522°N 6.640°E |
| | | | AC012 | 18854 | | |

calibration models when addressing cross-sensitivity issues arising from $O_3$. For comparison purposes, the same assessment is conducted utilizing $O_3$ measurements collected from the co-location reference stations (ZUE and LAU).

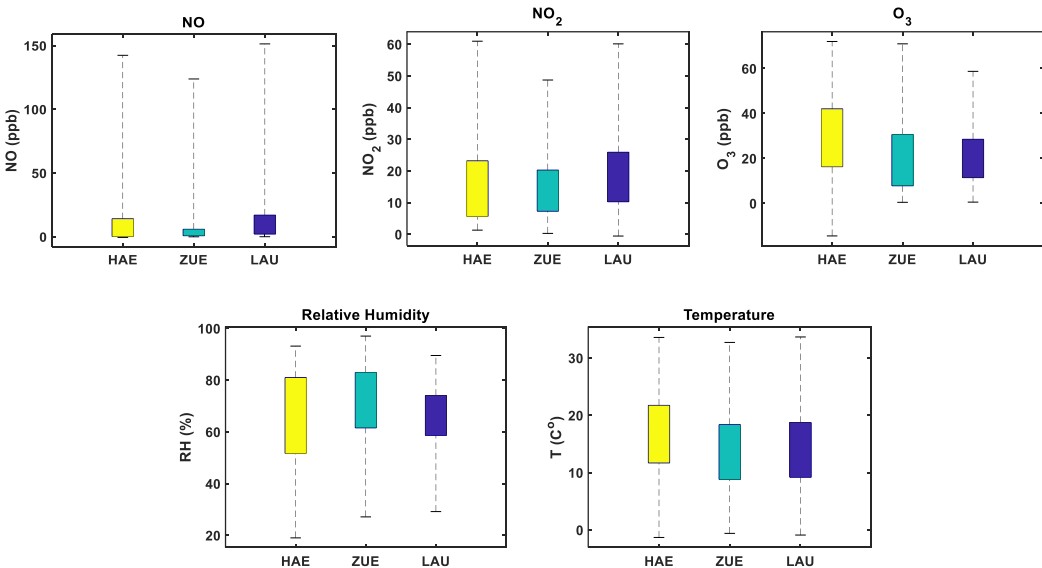

**Figure 1.** Box plots showing meteorological variables and pollutants concentrations at deployment sites using 10 min averaged data. The central line indicates the median, and the bottom and top edges of the box indicate the 25th and 75th percentiles of the data, respectively. The whiskers extend to the minimum and maximum values.

## 2.3 Calibration

The application of calibration transfer methods may facilitate the effort needed to obtain valuable measurements of air pollutants from low-cost sensors. Fig. 3 illustrates the two cases investigated by this study to examine the transferability of calibration





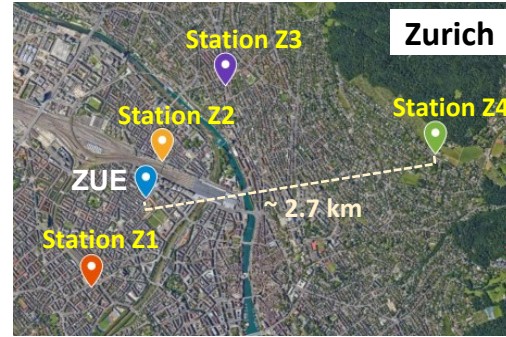
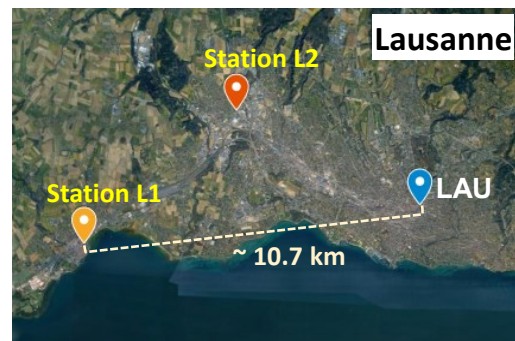

**Figure 2.** Map view of low-cost SUs co-location sites featuring nearby monitoring stations, in both Zurich and Lausanne (© Google Earth 2023).

between different (but identical) SUs. For Case A, each global calibration model was trained on a dataset from one SU, denoted as (Primary SU), and then applied to the rest of the SUs, denoted as (Secondary SUs), available at the same location. This case is designed to examine the ideal scenario and to serve as a benchmark. For Case B, each global calibration model was applied to Secondary SUs installed at different sites than the Primary SU. Every SU was once a Primary SU in both cases.

Calibration transfer approach is advantageous in networks consisting of a significant number of low-cost SUs. Instead of individually characterizing and calibrating each SU, it may suffice to characterize and calibrate a representative SU or a subset of units and then apply the acquired global calibration models to the remaining units within a network of low-cost SUs. These models can also be applied to other SUs of the same type, both those in close proximity to the calibrated units (Primary SUs) (e.g., same city, similar emission conditions) and those further away (e.g., same city, differing emission conditions).

Our calibration strategy (illustrated in Fig. 3) is designed to enhance model performance by minimizing cross-sensitivity variance and sensor-to-sensor variability. As $O_3$ could cause significant interference to $NO_2$ low-cost sensors, $O_3$ measurements were included in the features set of the calibration models.

### 2.3.1 Data investigation and preparation

Evaluating the consistency of SUs is recommended to determine whether similar electrochemical sensors respond to target changes similarly (Giordano et al., 2021). Higher consistency and reduced error sources such as sensor-to-sensor variations would pave the way for optimum transferability of calibration. Therefore, consistency was mainly assessed and addressed by: 1. Pairwise Pearson correlation ($R$) between identical low-cost sensors deployed at the same site, as shown in Table 2, where the results indicate significant correlations between the low-cost sensors. 2. Pearson correlation ($R$) between low-cost sensors and their corresponding reference measurements (Table 3). 3. Standardization of features, because identical sensors may have different baseline levels, even if coming from the same manufacturer and deployed at the same location, as shown in Fig. 4. Therefore, to tackle this issue, standardization (Z-score) was applied, in which all features have a mean of zero ($\mu = 0$) and one standard deviation ($\sigma = 1$). This results in almost completely uniform signals from the electrochemical sensors, across all SUs,





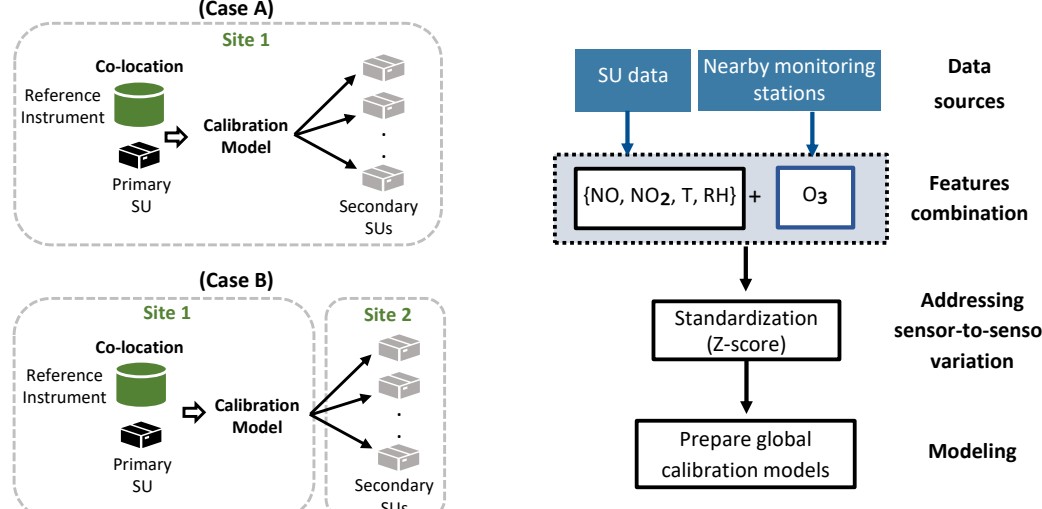

**Figure 3.** Scheme of the two cases of calibration transfer between different SUs (left), and the architecture of the global calibration model (right).

**Table 2.** Pairwise Pearson correlation ($R$) between the electrochemical sensors of different SUs.

| Site | SU - SU | | $NO_2$ | | NO | |
|---|---|---|---|---|---|---|
| | | | NO2_A | NO2_B | NO_A | NO_B |
| HAE | SU009 | SU010 | 0.83 | 0.96 | 0.96 | 0.74 |
| | SU009 | SU011 | 0.94 | 0.95 | 0.99 | 0.97 |
| | SU009 | SU012 | 0.97 | 0.97 | 0.97 | 0.92 |
| | SU010 | SU011 | 0.94 | 0.95 | 0.95 | 0.63 |
| | SU010 | SU012 | 0.88 | 0.95 | 0.97 | 0.91 |
| | SU011 | SU012 | 0.98 | 0.98 | 0.97 | 0.88 |
| ZUE | SU009 | SU010 | 0.91 | 0.96 | 0.98 | 0.81 |
| LAU | SU011 | SU012 | 0.97 | 0.98 | 0.98 | 0.84 |

185 especially when exposed to similar environmental conditions. Overall, this reduces the sensor-to-sensor variations, making it possible for global calibration reproducibility.

$O_3$ measurements acquired from the nearby monitoring stations are available in 1 h resolution, therefore, calibration models in this study are trained and tested based on 1 h data. When training a model (with Primary SU data), $O_3$ measurements were obtained from the co-location reference stations (either ZUE or LAU). When testing the global models (with Secondary SU

190 data): 1. For Case A, $O_3$ measurements were obtained from the reference station within the NABEL network (either HAE, ZUE or LAU), since the Secondary SUs are located at the same co-location site as Primary SUs. 2. For Case B, $O_3$ measurements





**Table 3.** Pearson correlation ($R$) between electrochemical sensors of SUs and their corresponding reference instruments.

| Site | SU | NO₂ | | NO | |
| --- | --- | --- | --- | --- | --- |
| | | NO2_A | NO2_B | NO_A | NO_B |
| HAE | SU009 | 0.44 | 0.68 | 0.84 | 0.80 |
| | SU010 | 0.74 | 0.60 | 0.81 | 0.72 |
| | SU011 | 0.69 | 0.83 | 0.87 | 0.81 |
| | SU012 | 0.57 | 0.77 | 0.88 | 0.85 |
| ZUE | SU009 | 0.62 | 0.77 | 0.83 | 0.87 |
| | SU010 | 0.86 | 0.69 | 0.84 | 0.74 |
| LAU | SU011 | 0.74 | 0.84 | 0.85 | 0.81 |
| | SU012 | 0.80 | 0.88 | 0.91 | 0.84 |

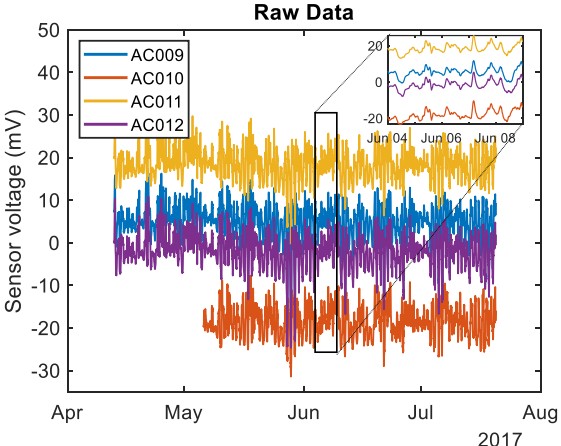
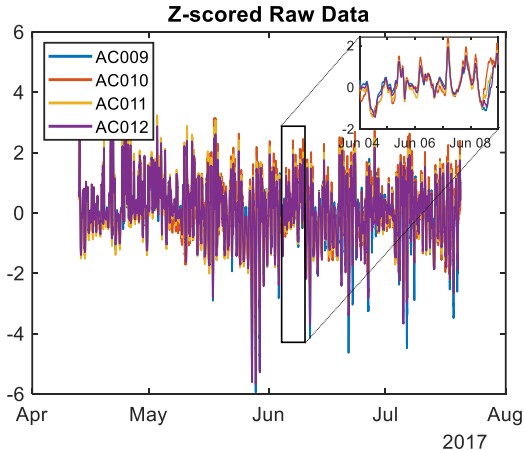

**Figure 4.** A comparison of raw NO2_A measurements before and after Z-score application, for each SU deployed at HAE.

were obtained from the nearby monitoring stations, replicating a real-world scenario in which Secondary SUs are installed at a different location without being collocated with reference instruments.

O₃ measurements obtained from the co-location reference sites and other nearby monitoring stations throughout the entire year 2017 were analyzed, in an effort to examine the consistency of O₃ concentrations among these stations (see Fig. 5). Analyzing the daily average of these measurements revealed a strong correlation between the co-location reference station and all nearby stations, in both Zurich and Lausanne, as depicted in the inset tables of Fig. 5. This indicates a consistent variability of O₃ across all stations in each site, implying that the positive contribution of any nearby station would enhance the model's capability to capture O₃ cross-sensitivity.





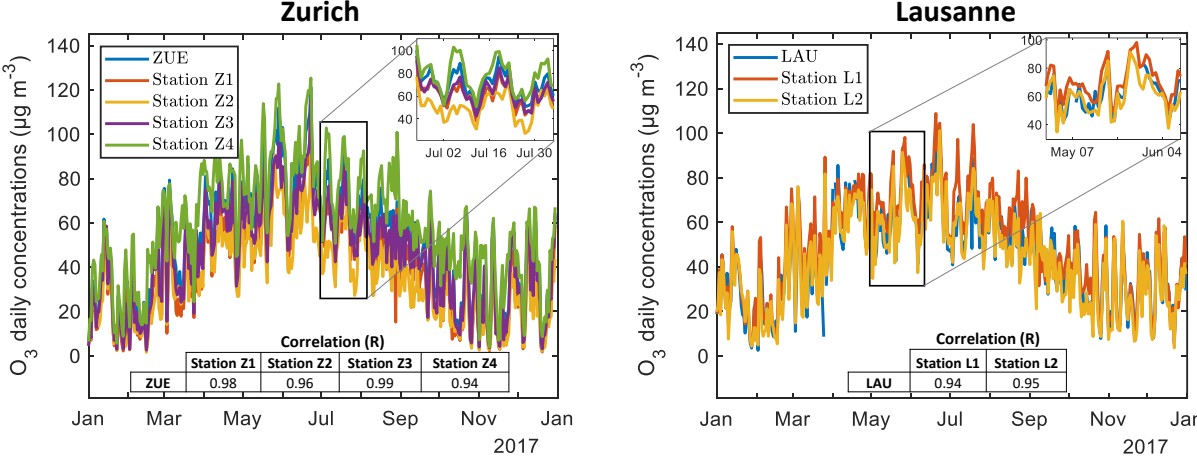

**Figure 5.** Daily $O_3$ measurements from the co-location reference station and the other nearby stations in both Zurich and Lausanne. The inset tables list the Pearson correlation ($R$) between each nearby station and the co-location reference stations.

### 2.3.2 ML-based calibration transfer method

In this study, three different ML-based calibration algorithms were used: Multivariate Linear Regression (MLR), Support Vector Regression (SVR), and Random Forest (RF). These algorithms were employed to estimate atmospheric concentrations of $NO_2$ and $NO$ based on a set of features (predictors). The choice of ML algorithms and features followed the approach by Bigi et al. (2018), as we utilized the same dataset. For the training of global calibration models for $NO_2$ and $NO$, six features were initially used: voltage signals of the four electrochemical sensors: NO_A, NO_B, NO2_A, and NO2_B, temperature and relative humidity. Additionally, our proposal suggests incorporating $O_3$ obtained from nearby monitoring stations. To evaluate the influence of incorporating $O_3$ on global models' performance, two sets of models were formulated and assessed. One set exclusively relied on SU data as features, while the other integrated $O_3$ into the feature set. Following this, a comparative analysis was carried out.

The models were trained and analyzed in MATLAB utilizing the `fitlm()` function for MLR, the `LIBsvm` software package for SVR, and the `TreeBagger()` function for RF. A K-fold cross-validation approach was used to address overfitting, where the training dataset (Primary SU) was divided into five folds (blocks) as depicted in Fig. 6. Here, we chose $k = 5$ based on the recommendation by Rodriguez et al. (2009). One block (20 % of the dataset) was used for validation, and the remaining blocks (80 % of the dataset) were used for training. This process was repeated five times (5 parts). In each split process, the block sampling approach introduced by Schultz et al. (2021) was followed to avoid the spurious correlation between training and validation sets. A grid search was applied to find the best hyperparameters, which were subsequently used to train the entire training set. The model was then evaluated using a test dataset (Secondary SUs). In RF models, the variable (predictor) importance can be calculated by randomly permuting each variable in the decision tree and averaging the estimation error over the forest (Breiman, 2001). The importance of a variable to the model increases as the estimation error increases.





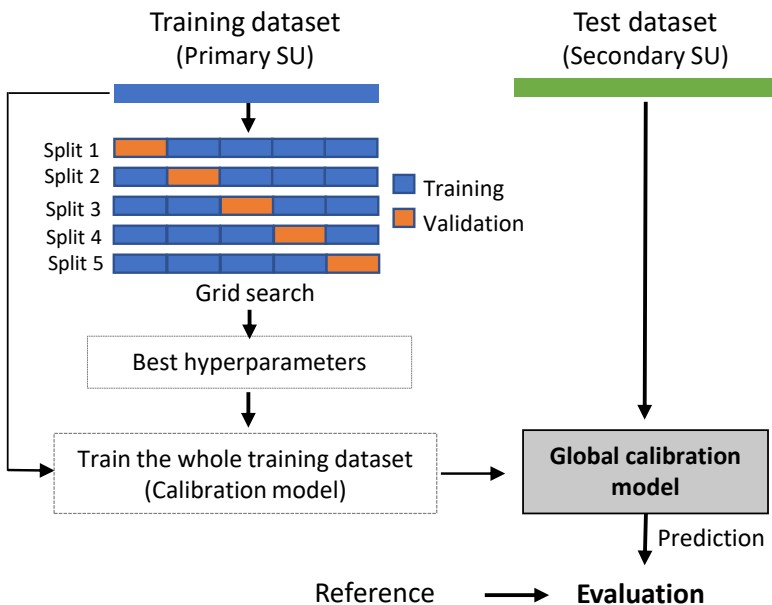

**Figure 6.** An illustration of the k-fold cross-validation approach.

## 3   Results and Discussion

### 3.1   Performance of calibration transfer

The global calibration models were evaluated using various goodness-of-fit metrics (MAE, $R^2$, and RMSE), see Appendix A. Fig. 7 summarizes the overall evaluation results for the two sets of global calibration models (with and without $O_3$) for $NO_2$ and NO in both cases. The results presented here and throughout this paper are based on the $O_3$ measurements obtained from "Station Z1" in Zurich and "Station L1" in Lausanne (Fig. 2). The results indicate successful transferability of the calibration models across SUs for $NO_2$ and NO, with Case A showing superior performance compared to Case B. In Case A, errors between the Primary and Secondary SUs are minimal, primarily because both SUs share the same environmental characteristics. Thus, Case A has a higher level of transferability than Case B. Moreover, the results indicate that, on average, RF consistently outperforms MLR and SVR, which aligns with the conclusions from Bigi et al. (2018) investigated individual calibration models using the same dataset. The major outcome from this study is that global calibration models perform better when including nearby monitoring stations' $O_3$ measurements in the feature set. In Case A, the RF-based $NO_2$ models demonstrated their highest transferability performance with an $R^2$ of 0.96 and an RMSE of 2.0. The corresponding averages were $0.90 \pm 0.05$ for $R^2$ and $3.4 \pm 0.9$ ppb for RMSE. In contrast, Case B exhibited a different performance profile, with the best $R^2$ value being 0.76 and an RMSE of 5.0. The averages in Case B were $0.65 \pm 0.08$ for $R^2$ and $5.5 \pm 0.4$ ppb for RMSE. Comparing NO models to $NO_2$ models, the former displayed superior transferability. In Case A, the RF-based NO models achieved an impressive $R^2$ value of 0.99 and an RMSE of 1.6 ppb, along with averages of $0.97 \pm 0.02$ for $R^2$ and $3.1 \pm 0.8$ ppb for RMSE.





In Case B, the best performance for NO models was characterized by an $R^2$ of 0.87 and an RMSE of 5.0 ppb, with averages of $0.82 \pm 0.05$ for $R^2$ and $5.8 \pm 0.8$ ppb for RMSE. Generally, NO models show better transferability than $NO_2$ models. Further details can be found in Tables S1 through S6 in the supplement. In comparison with existing literature such as Vikram et al.
240 (2019); Wang et al. (2023), these results demonstrate notable advancements.

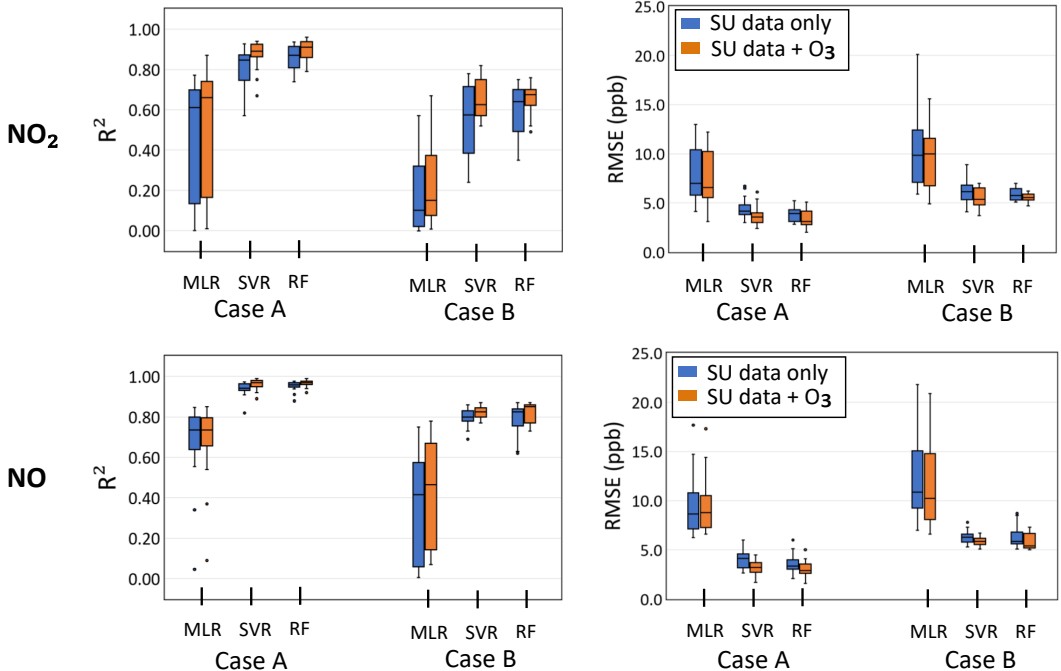

**Figure 7.** Average results of evaluating the performance of global calibration models for NO and $NO_2$, based on MLR, SVR, and RF techniques for both cases A and B. $O_3$ measurements were obtained from "Station Z1" in Zurich and "Station L1" in Lausanne.

The inclusion of $O_3$ measurements has resulted in noteworthy enhancements in predictive accuracy and generalizability, as indicated by the increased $R^2$ values and reduced RMSE values, particularly pronounced in the SVR and RF models. To comprehensively assess the impact on the global models, we explored the incorporation of $O_3$ measurements from all nearby monitoring stations in Zurich and Lausanne. Interestingly, every station contributed positively to model performance. Fig. 8
245 reports the average enhancements (%) in $R^2$ and RMSE for $NO_2$ RF global models by each nearby station in comparison with the co-location reference station, in both Zurich and Lausanne.

To better understand this interesting finding, we examined each global model's performance in terms of RMSE (%), as shown in Fig. 9. In Case A, notable enhancements in the performance of all RF-based models for $NO_2$ and NO were observed, with the $NO_2$ models experiencing a substantial improvement of up to 42 % and the NO models showing an improvement of
250 up to 25 %. In contrast, Case B demonstrated more pronounced improvements when Secondary SUs were located at ZUE. The RF-based models exhibited an enhancement of up to 17 % and 21 % for $NO_2$ and NO, respectively. Interestingly, no significant improvement was observed when the Secondary SUs were located at LAU. This finding can be attributed to higher $O_3$ levels



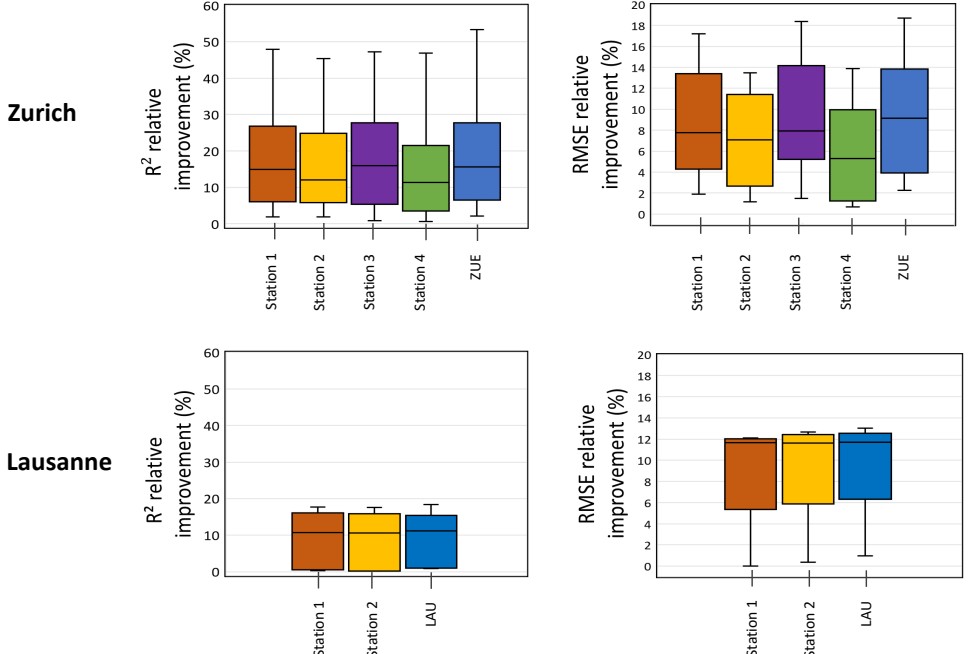

**Figure 8.** The $R^2$ and RMSE average positive improvements (%) of global $NO_2$ RF models in Case B, when including $O_3$ measurements from each nearby station in comparison with the co-location reference station in Zurich and Lausanne.

in ZUE (background site) compared to LAU (traffic site) (Fig. 1), leading to increased cross-sensitivity of low-cost sensors in ZUE. Consequently, the inclusion of $O_3$ measurements allowed the models to effectively capture and account for its influence, resulting in improved prediction accuracy. The feature importance plots (Fig. 10) provide further proof by indicating a higher significance of $O_3$ in the $NO_2$ and NO models of ZUE compared to LAU, thereby reinforcing its key role in capturing model variations.

The results also reveal that the transfer of $NO_2$ and NO calibration models to SU (AC010) resulted in the lowest performance among all Secondary SUs. Table 3 provides insights into the potential reasons behind this outcome, showing that for SU (AC010), NO2_A exhibits a stronger correlation with the reference $NO_2$ measurements compared to NO2_B. Furthermore, the feature importance plots (Fig. 10) indicate that NO2_A has a more significant influence on predicting $NO_2$ than NO2_B for models trained with the Primary SU (AC010), which is the opposite for the rest of the calibration models. Thus, we infer that the discrepancies in the correlation between counterpart features in the training and test datasets substantially impact the calibration transfer between SUs of the same make. Higher disparities suggest that the model may not generalize well to new data, which raises concerns about its overall performance. Accordingly, when selecting a Primary SU for the final global calibration model, it is crucial to select a SU that demonstrates representative feature importance for the other SUs to which the model will be transferred.



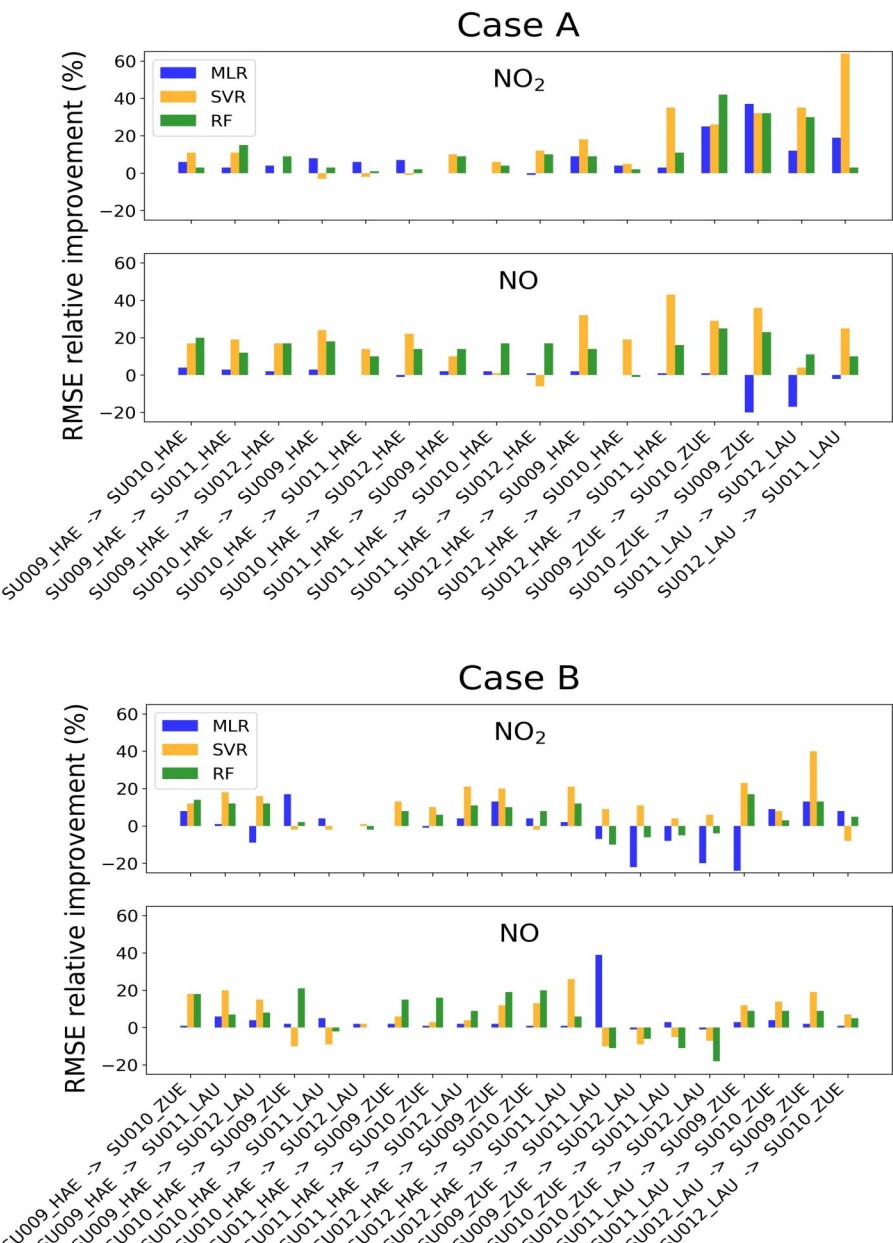

**Figure 9.** An illustration of the performance enhancement achieved by incorporating $O_3$ for both Cases A and B. X-axis represents (Primary SU -> Secondary SU).

In some cases, the poor performance of ML-based calibration models can be attributed to the nature of ML algorithms. As an example, many meteorological variables exhibit periodic variations and are correlated over time and space, with these corre-
270    lations changing with time. Unfortunately, ML algorithms are unaware of these relationships and have difficulty extrapolating

**NO₂**



**NO**

**Figure 10.** Feature importance plots for $NO_2$ and NO, for 1 h based measurements, including reference $O_3$.

periodic features correctly (Grover et al., 2015). Another possible reason is the existence of "unknown error sources", whose influences are not captured by ML models. As a result of the spatiotemporal difference between Primary and Secondary SUs, different external errors are imposed on ML models, which significantly impact their performance. Therefore, future solutions of such problems can be achieved by incorporating various measures such as feature engineering, which calculates derived properties that assist ML models in recognizing the more complex relationships imposed by various environmental conditions (Schultz et al., 2021; DeSouza et al., 2022).



## 3.2 Validity of the calibration transfer approach using a different dataset

Finally, in order to validate the reliability and effectiveness of our approach, we applied it to a different dataset collected in the town of Modena, in the Po valley, an European air pollution hotspot. The dataset was described and investigated in a previous work by Baruah et al. (2023). This allowed us to assess its robustness in diverse scenarios and identify the conditions necessary for a successful implementation. The Modena dataset consists of measurements obtained from twelve SUs deployed in Modena, Italy. Two different sites were selected for the co-location of these SUs with reference stations: an urban-background site, where $NO_2$, NO and $O_3$ reference measurements are available, and an urban-traffic site , where only NO and $NO_2$ reference measurements are available. Figs. S1 and S2 in the supplement, illustrate the temporal deployment of the Modena SUs and pollutants concentrations measured by the reference instruments. The deployment periods are sparsely distributed and span a period of approximately twenty months. Modena SUs were deployed for the shortest period of time at the urban-traffic site; some were deployed for around two weeks. This dataset can be used to validate our calibration transfer method. Modena SUs are equipped with three electrochemical sensors: $NO_2$ (Alphasense NO2-B43F), NO (Alphasense NO-B4), and OX (Alphasense OX-B431), as well as temperature and relative humidity sensors. According to our calibration strategy, since that OX sensor is available, it will be utilized as a source of $O_3$ data. This dataset has been analyzed, and the best features combination was identified, as stated in Eq. (1).

$$NO_2 = \text{function}(NO2\_we, NO2\_aux, NO\_we, NO\_aux, OX\_we, OX\_aux, T, RH)$$
$$NO = \text{function}(NO2\_we, NO2\_aux, NO\_we, NO\_aux, OX\_we, OX\_aux, T, RH)$$

(1)

The correlation analysis was explored (see Table S7 in the supplement). According to these investigations, all NO low-cost sensors and some $NO_2$ and OX low-cost sensors have a very low correlation with their corresponding reference measurements in the urban-background site. Fig. 11 shows results of the overall calibration transfer performance of $NO_2$ and NO models for the two sites. The findings of these results can be summarized as follows: 1. There is consistency with the results from the Switzerland dataset, in which RF outperforms MLR and SVR, and calibration transfer within the same site (Case A) achieves better performance than in Case B. Also, NO models show better transferability than $NO_2$ models. 2. It is possible that some models were unable to be transferred, presumably due to low correlation (pairwise and with their corresponding reference measurements), which is more prominent in NO low-cost sensors at the urban-background site. Moreover, the sparse deployment of SUs in the urban-background site and the short colocation period in urban-traffic can affect the generalizability of global models. 3. Despite the urban-traffic measurements having a short colocation period compared to urban-background measurements, the calibration transfer of urban-traffic data performed better than that of urban-background measurements, especially in Case B.

Based on our analysis of the Modena dataset, it is evident that three main conditions are required for the proposed calibration protocol to provide the best transferability of calibration models: 1. High correlation (pairwise, as well as with the reference measurements). 2. Sufficient period of colocation. 3. Using multiple electrochemical sensors dedicated to the same pollutant, such as the Switzerland dataset, which can enhance data reliability.





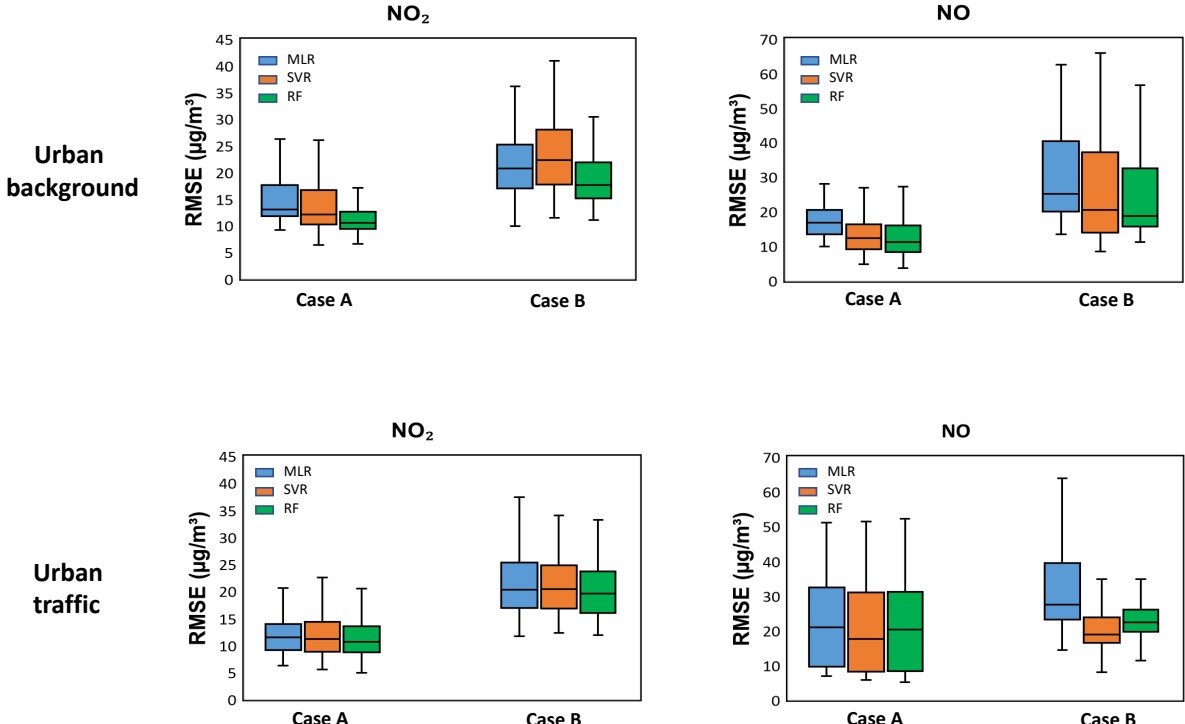

**Figure 11.** Average results of evaluating the performance of global calibration models for the Modena dataset at the urban background (UB) site and urban traffic (UT) site.

## 4 Conclusions

This study investigated the transferability of ML-based calibration models for $NO_2$ and NO across identical low-cost SUs deployed at similar and distant locations within Switzerland. Moreover, this study advocated enhancing global calibration models by incorporating $O_3$ measurements from available nearby monitoring stations. This strategic augmentation aims at effectively mitigating the cross-sensitivity issues associated with low-cost sensors in the absence of dedicated $O_3$ low-cost sensors (i.e., OX sensors), which is expected to improve the model's performance. The results of this study showed excellent

calibration transferability between SUs located at the same site (Case A), with the average performance of RF-based models being $R^2 = 0.90 \pm 0.05$ and RMSE = $3.4 \pm 0.9$ ppb for $NO_2$, and $R^2 = 0.97 \pm 0.02$ and RMSE = $3.1 \pm 0.8$ ppb for NO. The results also showed good transferability between SUs deployed at distant locations (Case B), which resulted in an average performance of $R^2 = 0.65 \pm 0.08$ and RMSE = $5.5 \pm 0.4$ ppb for $NO_2$, and $R^2 = 0.82 \pm 0.05$ and RMSE = $5.8 \pm 0.8$ ppb for NO. These results reveal notable advancements compared to the existing literature.

Our research identifies some key conditions for optimal global calibration model performance. These conditions are: 1. Robust correlation between sensors and their corresponding reference stations. 2. Similar pollutant levels are present at both





the Primary and Secondary SU locations, as some machine learning algorithms cannot extrapolate beyond the range of training data. 3. The utilization of multiple electrochemical cells within each SU targeting the same pollutant to enhance data reliability.

To conclude, the outcomes of our study will provide novel insights into the capability of ML models to generalize calibration models and emphasize the importance of utilizing publicly available data sources in order to improve the reliability of low-cost air quality sensors.





**Appendix A: Evaluation metrics and raw results of calibration transfer approach**

Three parameters were used to evaluate the overall performance of the calibration performance: $R^2$, RMSE, and Mean Absolute Error (MAE), given in Eqs. (A1)-(A3), respectively (Jolliff et al., 2009).

$$R^2 = 1 - \frac{\sum_{i=1}^{n}(y_i - \hat{y}_i)^2}{\sum_{i=1}^{n}(y_i - \bar{y}_i)^2} \tag{A1}$$

$$RMSE = \sqrt{\frac{1}{n}\sum_{i=1}^{n}(\hat{y}_i - y_i)^2} \tag{A2}$$

$$MAE = \frac{1}{n}\sum_{i=1}^{n}|\hat{y}_i - y_i| \tag{A3}$$

where $y$ denotes the reference measurements, $\hat{y}$ is the predicted values by the calibration model, and $\bar{y}$ is the mean of reference values. $R^2$ values range between 0 and 1, measuring how much the independent variables (features) can explain

the variation in the dependent variable (i.e. reference measurements). RMSE and MAE quantify the deviation between the calibrated values and their corresponding reference values.



*Data availability.* All raw data can be provided by the authors upon request.

*Competing interests.* The authors declare that they have no conflict of interest.

*Acknowledgements.* TUM authors wish to express their thanks to their funders: the German Academic Exchange Service (DAAD), and
the Institute for Advanced Study at the Technical University of Munich. Alessandro Bigi acknowledges funding from the European Union
NextGenerationEU program.

*Financial support.* TUM authors are supported by the German Academic Exchange Service (DAAD)(grant no. 57552340), and the Institute
for Advanced Study at the Technical University of Munich (grant no. 291763). Alessandro Bigi is supported by "ECOSISTER" project (grant
no. CUP E93C22001100001), funded by the European Union NextGenerationEU program, under the National Recovery and Resilience Plan
(NRRP) Mission 4 Component 2 Investment Line 1.5..



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
