# Peer review of "Transferability of ML-based Global Calibration Models for NO2 and NO Low-Cost Sensors"

_Atmospheric Measurement Techniques, 2023_

## Author Comment (AC1)

**Response to Reviewers**

Dear Reviewers,

We are grateful for your comments and suggestions, which have helped us improve the manuscript. The necessary revisions have been implemented, which can be found in the attached file (highlighted in yellow). Below, we provide responses to your comments and suggestions, along with corresponding changes made in the revised manuscript, where applicable.

Sincerely,

On behalf of all authors,
Ayah Abu-Hani
* * *
**Response to Reviewer #1**

**General Comments**

*The manuscript presents research work looking at the application of machine learning (ML) techniques in developing a transferable calibration methodology for a network of low-cost sensor units (SU) focusing on NO and NO2 pollutants. This study assessed the performance for collocated and non-collated networks of low-cost sensors in different urban environments in Switzerland and Italy. They incorporated several commonly used variables (raw sensor signals, RH, temperature) in their algorithm but emphasised the key role ozone plays as an input in their model, concluding that the best results were obtained in studies involving co-located networks which have ozone as part of the input variable.*

Thank you for taking the time to review our manuscript. We do appreciate your feedback and have carefully considered your comments. We have improved the revised manuscript in response to your suggestions.

**Specific comments**

*The authors have used low-cost SU that have a pair of electrochemical sensors for the two species of interest (NO, NO2). The reviewer found it odd that the pair of signals were used in the model setup for each species. For instance in modelling the corrected NO2, both the 'NO2_A' and 'NO2_B' are used but I would expect these two signals to be very correlated as summarised in Table 2. I would have thought one of the pairs should be sufficient, particularly the one with the best R value in Table 3.*

The study by Bigi et al. (2018) demonstrated that the feature combinations with a pair of electrochemical sensors leads to better performance in calibration models compared to a single sensor. Given that we utilized the same dataset in our study, we maintained the same features for benchmarking purposes. Furthermore, Smith et al. (2019) reported the effectiveness of employing an

array of sensors rather than a single sensor. They utilized the instantaneous median signal from six identical electrochemical sensors for $NO_2$ and $O_3$, resulting in minimized randomized drifts and inter-sensor differences, thus, addressing some limitations of individual sensors.

We have updated the corresponding statements in our revised manuscript (P.16, L.310) and (P.18, L.331).

*While the reviewer agree with the authors on the inclusion of O3 for as input variable for the NO2 calibration (there are ample literature evidence for this), there are very little evidence for the same for NO, thus questioning if this could lead to overfitting/training dependence for NO on O3 and potentially result in additional error in transferability of this method to regions where O3 is high but low NOX.*

Thank you for pointing this out. Yes, we agree that limited evidence for $O_3$'s inclusion in NO calibration, so, we made sure to clarify this in our revised manuscript (P.4, L.125).

In this work, we've evaluated our method's transferability across diverse environmental conditions, including regions with high $O_3$ but low $NO_x$ levels. For example, training a calibration model on LAU data and transfer to ZUE showed improved performance (as depicted in Fig. 9 Case B), where ZUE is characterized with higher $O_3$ and lower $NO_x$ compared to LAU. This is also supported by the feature importance (Fig. 10), shows that in ZUE $O_3$ is among the most significant attributes.

Moreover, we've employed robust validation techniques to address concerns about overfitting and training dependence.

**Technical corrections**

*Figures 1 & S2, the caption describes the central line of box plots to mean the median but the median are not shown in these figures.*

The figures are now updated. We also included the mean values (indicated by "*") in the modified manuscript and Supplement. (Fig.1 & Fig.S2).

*P.11, line 232 & 234, units missing for the RMSE values. Autor need to correct instance of this in the whole manuscript*

Thank you. The manuscript has been modified accordingly (P.11, L.335 & L.337).

1. *17, line 311-312: the statement "Moreover, this study advocated enhancing global calibration models by incorporating $O_3$ measurements from available nearby monitoring stations." This statement is too generic, it implies that $O_3$ needs to be considered for all low-cost SU network calibration like CO, PM, $CO_2$ and NO (see reviewers' general comment about this species above) etc.*

   The statement is now modified in the updated manuscript, and it is made specific for $NO_2$ and NO (P.17, L.318).

2. *18 line 333, the statement "The utilization of multiple electrochemical cells within each SU targeting the same pollutant to enhance data reliability" needs to be revised in context of the reviewer's general comments above for: 1) NO species and 2) overfitting by incorporating pair sensor of reading for same species.*

Corresponding statements have been modified to address these comments in the updated manuscript (P.18, L.331) and (P.4, L.125).

**References**

Bigi, A., Mueller, M., Grange, S. K., Ghermandi, G., and Hueglin, C.: Performance of NO, NO 2 low cost sensors and three calibration approaches within a real world application, Atmospheric Measurement Techniques, 11, 3717–3735, https://doi.org/10.5194/amt-11-3717- 2018, 2018.

Smith, K. R., Edwards, P. M., Ivatt, P. D., Lee, J. D., Squires, F., Dai, C., Peltier, R. E., Evans, M. J., Sun, Y., and Lewis, A. C.: An improved low-power measurement of ambient NO2 and O3 combining electrochemical sensor clusters and machine learning, Atmos. Meas. Tech., 12, 1325–1336, https://doi.org/10.5194/amt12-1325-2019, 2019.

---

## Author Comment (AC2)

**Response to Reviewers**

Dear Reviewers,

We are grateful for your comments and suggestions, which have helped us improve the manuscript. The necessary revisions have been implemented, which can be found in the attached file (highlighted in yellow). Below, we provide responses to your comments and suggestions, along with corresponding changes made in the revised manuscript, where applicable.

Sincerely,

On behalf of all authors,
Ayah Abu-Hani
* * *
**Response to Reviewer #2**

*The manuscript presents a framework for the global machine learning (ML)-based calibration models for NO2 and NO electrochemical cells using data from low-cost sensor units (SUs) utilized in a previous study by Bigi et al. (2018). This study mainly focuses on calibration transferability among SUs when deployed at the same location (or with the same environmental conditions) and different locations (or with different environmental conditions), given that no explicit overlap exists between the training and testing data distributions. This approach uses a simple standardization to account for sensor-to-sensor variations. In addition, the author claims that a potential improvement in model transferability was achieved by using O3 from nearby regulatory air quality monitoring stations.*

Thank you for taking the time to review our paper. We do appreciate your feedback and have carefully considered your comments. The manuscript is revised accordingly.

**Minor comments:**

*Figures 1 and S2: Where are the central (median) lines in the Box plots? Please include the median line and mean (with a symbol) in the figures.*

The figures are now modified as in the updated manuscript and Supplement (Fig.1 and Fig.S2).

*Figure 4: What does the negative sensor voltage convey? No mention of this in the manuscript.*

Thank you very much for your remark regarding negative sensor voltages. The caption of Fig.4 is now modified to highlight this point with a brief statement.

A more detailed explanation can be as follows:

The type of low-cost sensors in our study are electrochemical cells (ECs) on electronic sensor boards. In presence of the target gas, ECs produce a small electric current which is approximately proportional to the concentration of the target gas. The electronic sensor board amplifies this signal and converts it to a voltage, which is then the raw signal that we process.

However, the current of the EC is also affected by other ambient parameters such as temperature and humidity, which can cause the sensor output to drift. At low concentrations of the target gas, this electric current can also be slightly negative. Therefore, the electronic sensor board applies an electronic zero offset to the signal to ensure always a positive sensor output voltage.

The auxiliary electrode is affected by ambient parameters in the same way as the working electrode, however, it is not affected by the target gas, as it is not exposed to it.

An electronic zero offset is added to both working electrode and auxiliary electrode. Please note, that these electronic zero offsets are independent of each other, i.e. they are likely to be slightly different. If the zero offset of AE is significantly higher than the zero offset of WE, then WE-AE will be constantly negative. Furthermore, the chemical activity on auxiliary and working electrode may be different from each other, which could also lead to a negative sensor signal (WE-AE).

A more complete formula to calculate a compensated sensor signal would actually be (WE – WE_0) – (AE – AE_0) (or parametrized variations of this). However, as WE_0 and AE_0 are constants, applying "Z-score" after performing WE-AE, make WE_0 and AE_0 unimportant to apply.

***Appendix A and Line 222: Although MAE was mentioned as one of the measures for quantifying the deviation between the calibrated values and their corresponding reference values, it was never discussed in the main manuscript. Tables S1-S6 and Figures S3-S6 are not referred to in the main manuscript.***

Thank you for pointing this out. Our discussion is focusing on $R^2$ and RMSE, so, we have modified the manuscript accordingly (P.11, L.225). However, MAE results still exist in tables in the appendix in case they might be for interested researchers.

The manuscript is modified to refer to Figures S3-S6 (P.16, L.299) and Tables S1-S6 (P.12, L.241).

***Line 232: RMSE units are missing.***

Units are now added in the updated manuscript (P.11, L.335 & L.337).

***Figure 9: In the caption, please include how the RMSE relative improvement (%) was estimated.***
Thank you for the suggestion. The relative improvement was computed using the formula: [(new−old) / old × 100%]. The caption in the manuscript is modified accordingly (P.14, caption of Fig.9).

**References**

Bigi, A., Mueller, M., Grange, S. K., Ghermandi, G., and Hueglin, C.: Performance of NO, NO 2 low cost sensors and three calibration approaches within a real world application, Atmospheric Measurement Techniques, 11, 3717–3735, https://doi.org/10.5194/amt-11-3717- 2018, 2018.

Smith, K. R., Edwards, P. M., Ivatt, P. D., Lee, J. D., Squires, F., Dai, C., Peltier, R. E., Evans, M. J., Sun, Y., and Lewis, A. C.: An improved low-power measurement of ambient NO2 and O3 combining electrochemical sensor clusters and machine learning, Atmos. Meas. Tech., 12, 1325–1336, https://doi.org/10.5194/amt12-1325-2019, 2019.